# Estimation of extreme wave heights return period from short-term interpolation of multi-mission satellite data: application to the South Atlantic

Julio Salcedo-Castro[1], Natália Pillar da Silva[2], Ricardo de Camargo[2], Eduardo Marone[3], and Héctor H. Sepúlveda[4]

[1]Centro de Estudios Avanzados, Universidad de Playa Ancha, Traslaviña 450, Viña del Mar, Chile
[2]Departamento de Ciências Atmosféricas, Instituto de Astronomia, Geofísica e Ciências Atmosféricas, Universidade de São Paulo. Rua do Matão, 1226, 05508-900, São Paulo, SP, Brasil
[3]Centro de Estudos do Mar, Universidade Federal do Paraná, Av. Beira Mar s/n, 83255-000, Pontal do Sul, PR, Brasil
[4]Departamento de Geofísica, Universidad de Concepción, Avenida Esteban Iturra s/n, Barrio Universitario, Concepción, Chile
*Correspondence to:* J. Salcedo-Castro (julio.salcedo@upla.cl)

**Abstract.** We analyzed the spatial pattern of wave extremes in the South Atlantic Ocean by using multiple altimeter platforms spanning the period 1993-2015. Unlike the traditional approach adopted by previous studies, consisting in computing the monthly mean, median or maximum values inside a bin of certain size, we tackled the problem with a different procedure in order to capture more information from short term events. All satellite tracks occurring during two-day temporal window were gathered in the whole area and then gridded data was generated onto a mesh size of $2° \times 2°$ through optimal interpolation. The Peaks Over Threshold (POT) method was applied, along with the generalized Pareto distribution (GPD). The results showed a spatial distribution comparable to previous studies and, on the other hand, this method allowed capturing more information on shorter time scales without compromising spatial coverage. A comparison with buoy observations demonstrated that this approach improves the representativeness of short-term events in an extreme events analysis.

# 1 Introduction

In the context of climate change, the study of extreme waves has special relevance, as changes on regional and global waves pattern will impact coastal infrastructure, sediment transport and beach erosion, especially in low elevation coastal areas (Mori et al., 2010; Hemer et al., 2012; Izaguirre et al., 2013). For instance, a 109 years numerical hindcast showed that the North Atlantic Ocean has experienced an increase of the yearly-mean significant wave height (SWH) north of 50°N (Bertin et al., 2013). These findings coincide with an increase in the global wind speed and significant wave height during the period 1985–2008, especially in the 90th and 99th percentiles (Young et al., 2011). Similar results were obtained by Chawla et al. (2013), who carried out a thirty one year wave hindcast (1979–2009).

In spite of the advantages of reanalysis and modeling for studying the wave climate and extreme wave heights, these methods require validation. It has been shown that ERA-40 reanalysis underestimates altimeter observations and has larger uncertainty when comparing wave and wind data (Caires and Sterl, 2003). On the other hand, Chu et al. (2004) validated SWH modeled results in South China Sea with observations from TOPEX/Poseidon (T/P) data. Similarly, Rocha et al. (2004) used altimeter data to validate a hindcast of the surface wave field in the South Atlantic Ocean. These authors mentioned that, given the limited local observational data, altimeter data represent an excellent source of information for the region. However calibration and validation of this information, along with inter-comparison, is necessary when dealing with mono-mission or multi-mission altimetry data (Zieger et al., 2009). Consequently, direct observations from buoys and altimetry remain the most reliable sources to study wave climate.

The study of wave extremes by using altimeter data has proved to be very reliable and useful (Zieger et al., 2009; Vinoth and Young, 2011; Young et al., 2011, 2012). However, the most severe limitation of satellite altimeter data is the temporal and spatial sparseness (Hemer et al., 2010). This has been solved by averaging satellite track data from multiple missions within a certain quadrangle. Carter (1993) computed monthly mean values of SWH in 2° latitude × 2° longitude bins and compared them with buoy data. Challenor et al. (2004) adopted a method similar to that of Carter (1993), representing satellite tracks by their respective medians within a 2° square. These authors used the 90th percentile to apply the Peaks Over Threshold (POT) technique and the generalized Pareto distribution (GPD) to obtain a map of the 50yr extreme wave height in the North Atlantic. Using a different approach, Chen et al. (2004) defined 1° square bins and computed the return values of extreme waves that were represented on a global map through optimal interpolation. In a different way, Wimmer et al. (2006) defined 2° squares to compute the median in each cell to obtain the return values. Vinoth and Young (2011) compared a few methods using different area sizes and representative statistics and concluded that, when using the POT method, a 2° square is a good approach because under-prediction and under-sampling are avoided.

Some limitations are described for the methods above presented. One of the difficulties of using altimeter data is the inability to capture the response to atmospheric storms systems of relatively small dimension that move faster than satellite tracks (Cooper and Forristall, 1997). This involves a decision about the grid size. In this respect, Tournadre and Ezraty (1990) carried out a statistical analysis and concluded that a coverage of up to 200 km still represents data being part of the same processes. In a similar exercise, Panchang et al. (1999) arrived at the same conclusions about the usefulness of satellite track data within an

area with a radius of 200 km. According to Shanas et al. (2014) the balance between spatial and temporal resolution is partially accomplished by using a multiplatform altimeter data, obtaining a mesoscale variability of 100 to 300 km. Other studies have shown that using the whole along track data instead of statistics is more consistent with local observations when comparing by means of distribution functions (Cooper and Forristall, 1997).

In summary, the approaches adopted so far in previous studies (Carter, 1993; Alves and Young, 2003; Challenor et al., 2004; Chen et al., 2004; Caires and Sterl, 2005; Wimmer et al., 2006) consist in computing the monthly mean, median or maxima from different satellite along track data and process this information in order to estimate the return period. In this study, we carried out an extreme wave analysis based on satellite altimeter data generated by a multi-satellite database that was already standardized, quality-controlled and corrected, from 1993 to 2015 (Queffeulou, 2004; Queffeulou and Croizé-Fillon, 2017).

The goal was to estimate the extreme value return based on data directly interpolated from along track satellite data instead of using spatially averaged data. This method would provide a balance between spatial and temporal coverage so that more information associated to short time scale (and mesoscale) processes can be preserved without affecting the independence and representation of events like storms besides obtaining a good representation of ocean wave climate.

## 2    Methods

### 2.1    Interpolation of along track satellite data

Satellite along-track altimeter data was obtained from the Laboratoire d'Océanographie Spatiale (LOS) and the Centre ERS d'Archivage et de Traitement (CERSAT), at the Institut Français de Recherche Exploitation de la mer (IFREMER) (France) (Queffeulou, 2004). As part of this project, altimeter measurements from 7 to 9 satellites are continuously quality-controlled, corrected and inter-calibrated to provide a homogeneous and consistent data set (Queffeulou, 2013; Queffeulou and Croizé-Fillon, 20 2017). The documentation for the global altimeter SWH dataset describes a screening procedure to eliminate spurious measurements without affecting extreme wave records (Queffeulou and Croizé-fillon, 2010). The process consists on eliminating highest values obtained by the application of a 100 km running window along track samples. Some criticism has arisen from some authors, arguing that this processing causes removal of extreme observations and therefore its application to analyze wave extremes is not reliable. However, a recent study by Hanafin et al. (2012) demonstrates that, after correction and process-25 ing, multi-mission altimeter data preserves extreme wave observations larger than 20 m during the passage of extra-tropical cyclones.

     The study domain is delimited by 0° S and 60° S latitude and 70° W to 25° E longitude, and the period was 1993–2015. As a way to preserve more information from short-term events (5 days or less), we adopted a different approach. Instead of computing monthly averages or medians as representative values within individual quadrants of a gridded field, we firstly 30 gathered all satellite tracks occurring during a two-day temporal window in the whole area and then generated gridded data onto a mesh size of 2° × 2° through optimal interpolation (objective mapping) (Fig. 1). Optimal interpolation is a standard and proven method consisting in a weighted linear combination of observations irregularly distributed (Wilkin et al., 2002; Melnichenko et al., 2014). Unlike smoothing methods, optimal interpolation is based on the data ensemble statistics, by ap-

plying the Gauss-Markov theorem. The objective is to assure a result where there is a minimum variance solution at each point. Therefore, provided there is a good coverage and knowledge of the data, this method yields quite accurate interpolated results (Bretherton et al., 1976). A detailed description of this method can be found in Daley (1991). The interpolation method was applied to grid scattered satellite tracks using a customized version of the OBJective MAPping interpolation (OBJMAP) package, developed by Kirill K. Pankratov, as part of a Mathworks® MATLAB external toolbox named 'Data Analysis and Modification' (datafun) provided by the Ocean Time Series Group at Scripps Institution of Oceanography (Pankratov, 1995).

The mesh size of 2° × 2° box was chosen because it was the most refined option to ensure that our 2-day averaged time series have no missing values in a >20 year range, from 1993–2015, and based on other authors (Cotton and Carter, 1994; Young, 1999; Woolf et al., 2002; Alves and Young, 2003; Hemer, 2010) and the results of Vinoth and Young (2011) about the acceptable tolerance of the peaks-over-threshold (POT) method to this grid resolution. A 2-days temporal window was shown to assure enough spatial coverage without compromising the temporal resolution and thus capturing most of the extreme events (short time scale) processes. This is a critical point as the main goal of this study is to try to capture as much information from extreme events as possible. A temporal window that is too long produces overlapping of many satellite tracks in a time scale where there is a mixing of different processes and, in some cases, some wave height extremes registered by some tracks can be masked by the following track. On the other hand, a temporal window that is too short does not provide enough tracks to generate an acceptable interpolation and gridding in the study area. Therefore, a minimum window of 2-days seems to assure a better representation of extreme events without compromising spatial representativeness. Moreover, this temporal window of 48 hours allows fulfilling the condition of statistical independence and aleatory distribution of the events for each grid point (Palutikof et al., 1999).

Four time series of buoy data were obtained from the Brazilian National Buoys Program (PNBOIA) to compare the interpolated results. The location of each buoy, corresponding to Recife (RE), Santos (SA), Florianopolis (FL) and Rio Grande (RG), along with its distance to the nearest grid point and temporal coverage are shown in (Table 1).

**Table 1.** Buoys belonging to the Brazilian National Buoys Program (PNBOIA) used to compare the interpolated series from LOS.

| Buoy location | Latitude | Longitude | Distance to nearest grid point (km) | Temporal coverage |
|---|---|---|---|---|
| Recife (RE) | 34.58 S | 7.29 W | 102 | July 2012- March 2014 |
| Santos (SA) | 44.93 S | 25.27 W | 131 | January 2012 - April 2014 |
| Florianopolis (FL) | 47.39 S | 28.52 W | 89 | February 2011 - February 2013 |
| Rio Grande (RG) | 49.88 S | 31.58 W | 48 | April 2009 - January 2013 |

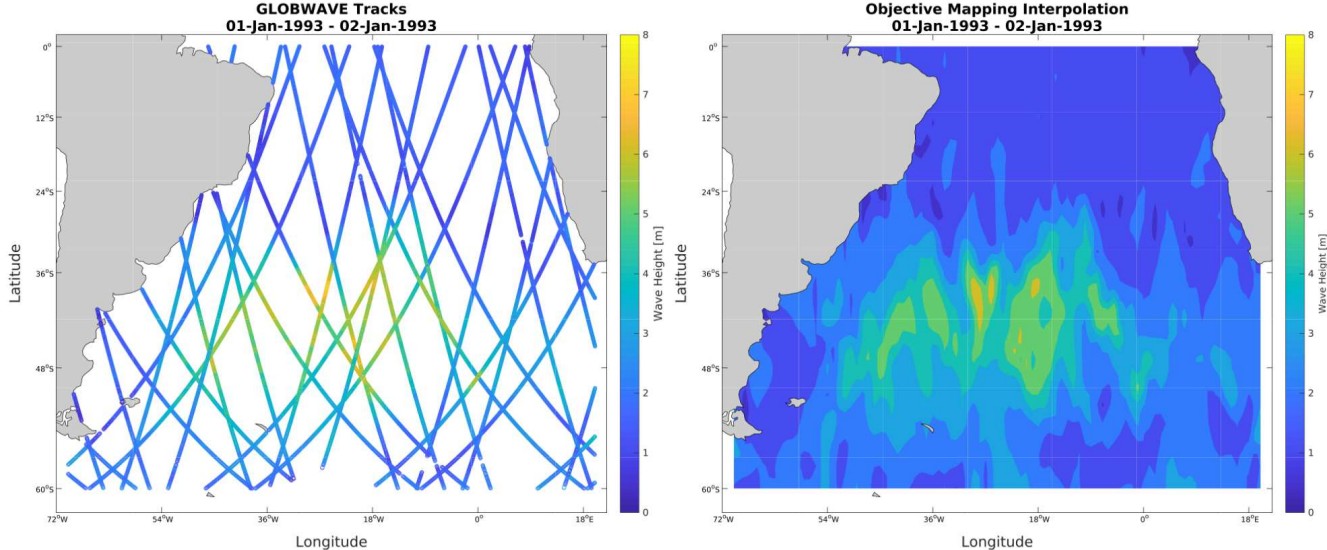

**Figure 1.** Example of multi-mission along-track data corresponding to a two-days temporal window (left) and $2° \times 2°$ gridded SWH field interpolated through objective mapping (right).

To assess the goodness of the results, the statistical parameters used were the bias, correlation coefficient (R), root mean square error (RMSE), and scattering index (SI), which were defined according to Shanas et al. (2014) as follows:

$$\text{BIAS} = \frac{1}{N} \sum_{l=1}^{N} (Ai - Bi) \tag{1a}$$

$$R = \frac{\sum_{l=1}^{N} [(Ai - \bar{A})(Bi - \bar{B})]}{\sqrt{\sum_{l=1}^{N} [(Ai - \bar{A})^2 (Bi - \bar{B})^2]}} \tag{1b}$$

$$\text{RMSE} = \sqrt{\frac{1}{N} \sum_{l=1}^{N} (Ai - Bi)^2} \tag{1c}$$

$$\text{SI} = \frac{\sqrt{\frac{1}{N} \sum_{l=1}^{N} [(Ai - \bar{A}) - (Bi - \bar{B})]^2}}{\bar{B}} \tag{1d}$$

The buoys time series were averaged every two days so as to obtain a representative value to compare with the single values obtained after interpolating the along-track altimeter data generated during the same period. Besides the statistics parameters described above, a quantile-quantile plot (Coles, 2001) was included to compare both series.

## 2.2 Peaks over threshold method

The application of the POT technique implies choosing a threshold value ($u$) over which the exceedances will be obtained. Therefore, this step is critical to adjust the exceedances to a GPD and needs to be carried out carefully. The choice of the threshold value was tested by using the mean excess plot (MEP) method (Ghosh and Resnick, 2010). This method evaluates how well the chosen threshold fits within an expected distribution of thresholds mean and these values are represented in the MEP as an approximately linear segment (Coles, 2001). We tested threshold values of 93%, 95% and 97% for all buoy time series.

The chosen method to estimate return values was the Generalized Pareto Distribution (GPD). The GPD has the following expression (Belitsky and Moreira, 2007):

$$G_{\xi,\beta}(y) = \begin{cases} 1 - \left(1 + \xi \frac{y}{\beta}\right)^{-\frac{1}{\xi}}, & \xi \neq 0 \\ \\ \left(1 - e^{-\frac{y}{\beta}}\right), & \xi = 0, \end{cases} \tag{2}$$

where $y$ represents the exceedances with respect to a threshold value $u$. Here, $\beta$ is a shape parameter and $\xi$ is a scale parameter. In this distribution, $y \geqslant 0$ for $\xi \geqslant 0$. The parameters $\beta$ and $\xi$ can be estimated by different methods, like the Maximum Likelihood Method (ML), Method of Moments (MOM), Pickand's Estimator (PKD) and the Probability Weighted Moments (PWM). From these methods, MOM and PWM exhibit better fit quality (Campos, 2009). Consequently, we adopted PWM to estimate the parameters of the GPD, as also recommended by Caires (2011). A Kolmogorov-Smirnov test was applied to test that each individual series of exceedances belongs to a GPD with the same parameters, with a 95% of significance. A two sample Kolmogorov-Smirnov test is a non-parametric hypothesis test that compares the sample's cumulative distribution functions and evaluates differences between each sample, indicating that those samples belong to similar distributions. In our study, random empirical time series were created using GPD shape and scale parameters for each grid point series and then compared with the excesses above threshold series.

To determine the extreme value for a certain return period by fitting the GPD, we used the following expression (Campos, 2009):

$$x_r = u + \frac{\beta}{\xi}\left[\left(\frac{n}{N_u}(1-p)\right)^{-\xi} - 1\right], \tag{3}$$

where $N_u$ is the total number of exceedances above the threshold, $x_r$ is the extreme to be computed, $n$ is the total number of observations in the series, and $p$ is the probability of not exceeding. The probability of not exceeding is given by:

$$p = 1 - \frac{1}{N_e}, \tag{4}$$

where $N_e$ is the total number of expected exceedances for a given return period $p_r$, obtained form the expression:

$$N_e = p_r \frac{N_u}{N_{years}}, \tag{5}$$

where $N_{years}$ is the number of years considered in the analysis. In this case, we computed the extreme value for a return period $p_r = 10$, $p_r = 25$ and $p_r = 50$.

The analytic procedures and computation of return values were performed with MATLAB *Wave Analysis for Fatigue and Oceanography* (WAFO) tool (Brodtkorb et al., 2000).

## 3   Results and discussion

An useful approach to evaluate the threshold choice for a POT analysis can be performed by the analysis of the mean excess plot. Fig. 2 shows the MEP for the GW data, both raw and optimally interpolated, calculated for the nearest point to the buoy data. There are differences between the raw and interpolated data (gray areas), but for all cases, all three tested values (93%, 95% and 97%) could be used as threshold value for the estimation of return period as they are located in a region with almost linear MEP (Fig. 2). Consequently, following Caires and Sterl (2005) and Challenor et al. (2004), we used the 97th percentile as threshold value, which was set as spatially variable (Alves and Young, 2003).

A reasonable agreement (R⩾0.64) is observed between the interpolated LOS data and the buoy series (Fig. 3), considering that the comparison is between two-days along-track altimeter data and two-days average buoy data. An increase from north to south in the magnitude of the BIAS and RMSE can be observed (Fig. 4). A similar trend in the average SWH is observed from north to south, where Recife shows an average SWH of 1.5 m (90th percentile: 1.9 m); Santos, 1.9 m (90th percentile: 2.7 m); Florianopolis, 2.1 m (90th percentile: 2.9 m), and Rio Grande, 2.3 m (90th percentile: 3.2 m). All these features and trends are well captured by the interpolated LOS series. Recife station shows a more regular wave climate caused by the uniform westward winds of the equatorial latitude, which is in contrast to the other stations more exposed to extreme waves associated to the passage of cyclonic events and ocean-swell from the south (Fig. 3). Rio Grande is the closest station to the coast (48 km) and there may be some corruption of the altimeter data by irregularities in the coastline can be responsible for this discrepancy.

In order to test the similitude between the optimally interpolated data with buoy measurements, the period from March to June of 2012 was selected, based on buoy data availability. These tracks (Fig. 5) were provided by the Group of Climate Studies of IAG/USP and were obtained with the methodology proposed by Murray and Simmonds (1991), applied to NCEP Reanalysis fields (Kalnay et al., 1996). The time series for the southernmost locations, Santos, Florianopolis and Rio Grande, are shown in Fig. 5b, where the passage of the selected cyclones is depicted. Rising values of SWH in all locations for cyclone cases A, B and I can be observed, which can be described as typical extra-tropical events affecting the whole southern coast of Brazil. Cyclones D, G and J can be characterized as subtropical cases that affect mainly the northernmost location (Santos) in terms of SWH. Intermediate situations like cyclones E and F appeared affecting only Rio Grande and Florianopolis. All these events showed better correspondence between buoy and optimally interpolated data at Santos station. Florianopolis presented

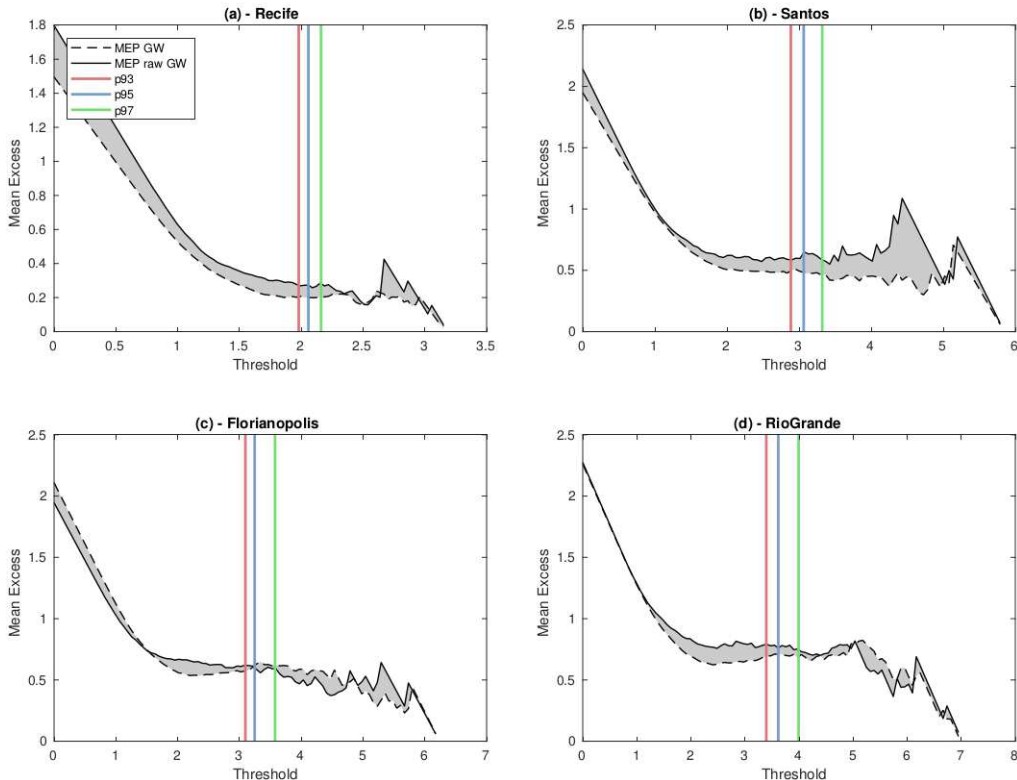

**Figure 2.** Comparison of mean excess plot (MEP) for Recife, Santos, Florianopolis and Rio Grande. The thresholds correspondent to 93, 95 and 97 percentiles (vertical lines) are shown.

an fluctuating behavior, whereas Rio Grande showed mostly a slight overestimation of optimally interpolated values. Therefore, in general, the short term variability is well captured by the optimally interpolated data, even with different behavior between the buoy locations.

Results of thresholds obtained after interpolation are shown in Fig. 6, along with results from ERA-Interim computed for the same period, in order to compare this approach with those results from gridded reanalysis products. Both distributions are similar and show the same general pattern, although data from LOS provides a more detailed description. However, these details should be treated with caution because additional analysis are necessary to demonstrate that optimal interpolation resolves more real structures in the data. Higher threshold values are observed south of 35°, exhibiting maximum values in the southeastern Atlantic, with values > 5 m. Similarly, Hemer et al. (2010) obtained monthly SWH climatology in the Southern Hemisphere and observed a strong latitudinal gradient, with largest wave heights toward the Indian Ocean. This is in agreement with the extreme values obtained in our study, where the highest wave height thresholds are shown nearby South Africa. Our

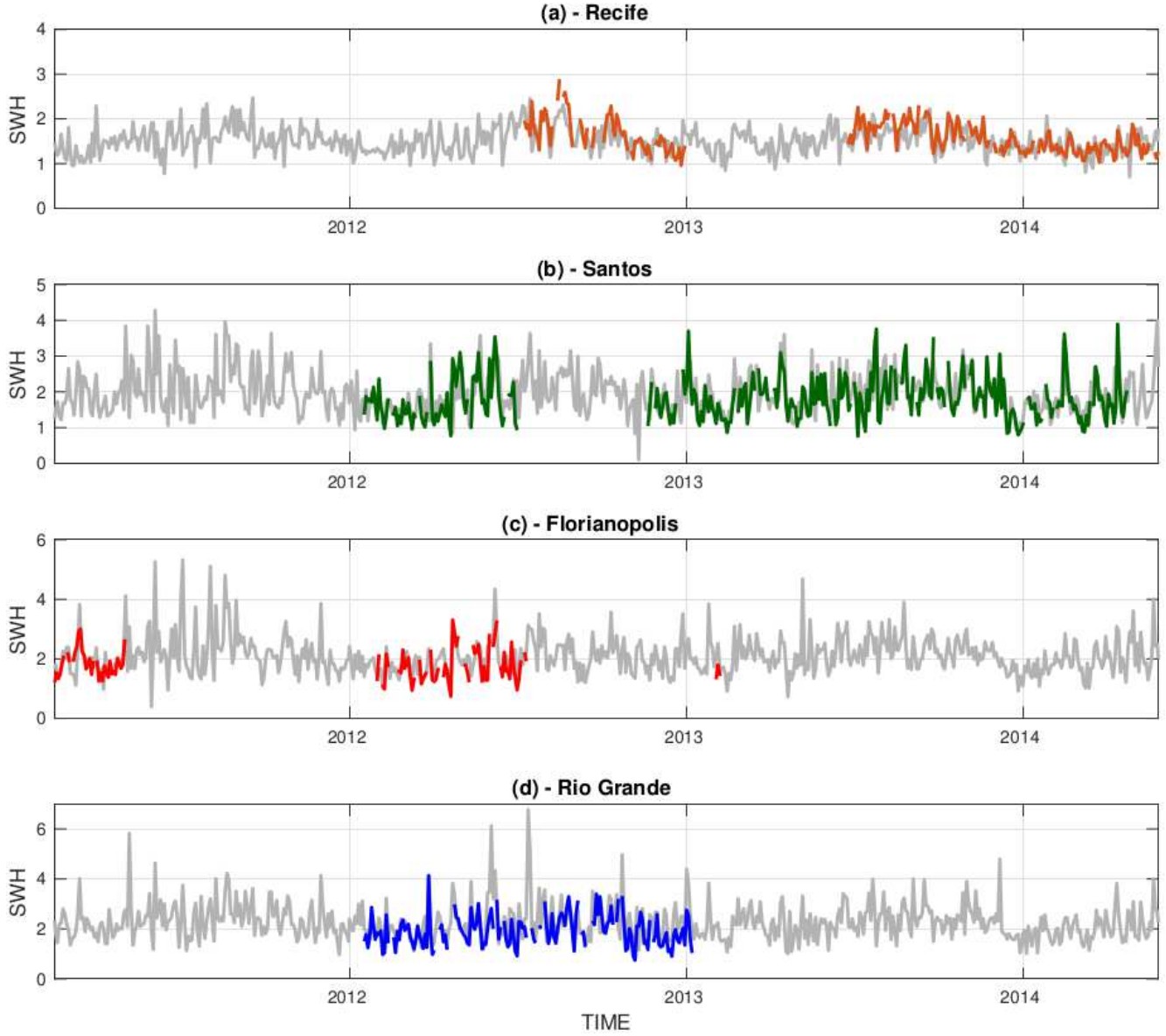

**Figure 3.** Time series for the LOS interpolated data (gray) in points near buoy stations in Recife (orange), Santos (green), Florianopolis (red) and Rio Grande (blue).

results are consistent with those described by other authors, like Mori et al. (2010), with SWH < 2 m in the equator and SWH > 6 m in the southwest region of the South Atlantic. The observed zonal variation of the extreme wave height agrees with that observed by Alves and Young (2003). In this sense, Alves and Young (2003) described three zonal bands: a higher latitude

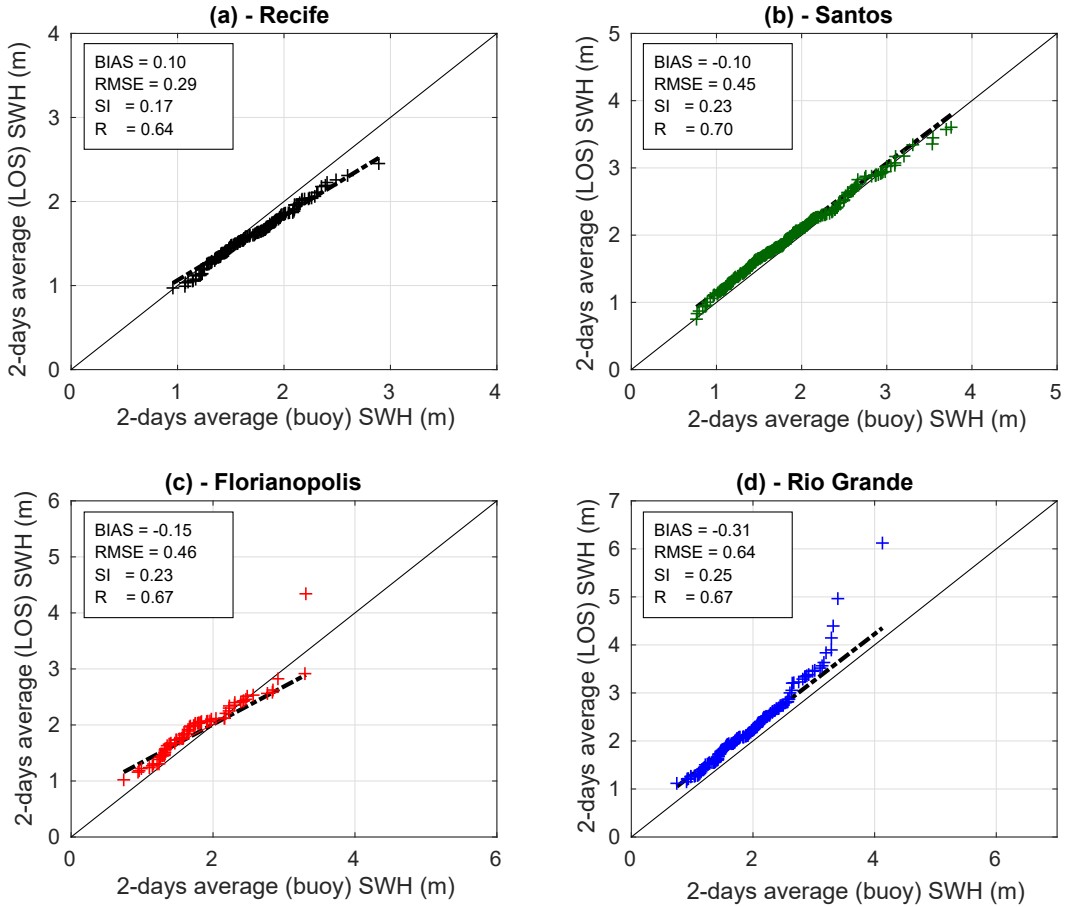

**Figure 4.** Comparison of QQPlots between 2-day averaged buoy data for stations in Recife (black-top left), Santos (green-top right), Florianopolis (red-bottom left), Rio Grande (blue-bottom right) and the nearest LOS interpolated time series.

region, characterized by higher SWH, an equatorial band, with lower SWH, and a transitional zone, between 20° and 40°. In this respect, Young et al. (2012) states that the higher SWH found in the sub-tropical region of the South Atlantic is associated to swell generated by storms in the Southern Ocean.

The 10yr, 25yr and 50yr return values are shown in Fig. 7, Fig. 8, and Fig. 9, respectively, including a comparison with
5  results from ERA-Interim. Our results are similar to those found by Young et al. (2012) and there is a close similitude to the zonal pattern and large extremes between 45° and 60° in the Southern ocean (Chen et al., 2004). However, our results showed a more detailed map, with distribution of return values 1-2 m lower (See Fig. 6a and 6b in Chen et al. (2004)). This difference can arise because Chen et al. (2004) used only one (TOPEX) altimeter data collected during a period of eight years. This contrast

has been highlighted by Shanas et al. (2014), who demonstrated that multi-mission products have advantages in estimating extreme waves when compared with mono-mission along-track data. Qualitatively, these results are similar to those obtained by Izaguirre et al. (2011), although these authors used monthly maximum SWH values. The extreme wave height distribution also resembles the description given by Jiang and Chen (2013), where a northeastward swell propagation from Drake Passage can

be distinguished in the subtropical area of the South Atlantic Ocean. In contrast, ERA-Interim produces higher return values for all the three periods analyzed. Our results agree with maximum return values around 13 m, whereas ERA-Interim overestimates this range, exhibiting maximum values of > 18 m, especially in the southeastern part of the domain. Other studies have demonstrated this ERA-I overestimation (Kumar and Naseef, 2015), as well as sub- and overestimation (Shanas and Sanil Kumar, 2014; Samayam et al., 2017), depending on local conditions and local processes. However, the main cause of this discrepancy

seems to be irregular distribution of tracks points in the domain, where, as consequence, the southeastern part has a lower spatio-temporal density of the tracks points. When plotting the mean difference between ERA-Interim and LOS return values (not shown here), it is possible to confirm that larger differences are associated to less data available in the region between 40° and the south boundary of the domain. This missing information causes an underestimation in that area but, in general, this method works well when data is available.

The more detailed pattern obtained in this study may be related to the higher temporal resolution that captures the passage of tropical and extra-tropical cyclones, in contrast to the results obtained by Caires and Sterl (2005). Another difference was given by the threshold of 97th percentile used in our study versus the 90th percentile used by Caires and Sterl (2005) and others. The better suitability of the 97th percentile would be related to the treatment of the along track data, which was not averaged in this study. On the other hand, the spatial variability of the return values resembles the description given by Caires and Sterl (2005),

who assert that a long data set and a chosen fixed threshold would assure a fair representation of major spatial features.

The chosen grid size produced results comparable to those obtained by other authors (Cotton and Carter, 1994; Young, 1999; Panchang et al., 1999; Woolf et al., 2002; Alves and Young, 2003; Vinoth and Young, 2011), confirming that a mesh element size of 2° × 2° is an acceptable choice. Furthermore, this size proved to be suitable to apply the POT method, which is sensitive to undersampling and threshold value, as compared to more robust methods like initial distribution method (IDM)

(Alves and Young, 2003; Vinoth and Young, 2011; Young et al., 2012). However, as stated by Vinoth and Young (2011) and Young et al. (2012), the availability of long time series allows acceptable results with the POT method. The application of POT and GPD to estimate extreme wave return values, consequently, is suitable to undertake this study, as also proven by previous work (Challenor et al., 2004; Wimmer et al., 2006).

An important point to consider is the coastal location of the buoys and the distance between the grid point and the coastal

buoy, as Shanas et al. (2014) mentioned in their comparison. These factors affect the results as buoys close to the coast are under complex interaction between the topography, waves and wind. On the other hand, satellite signal get contaminated due to the closeness to the continent and radiometer footprints.

## 4  Conclusions

A new methodological approach was proposed in this study. The multi-mission tracks were gathered in 2-days temporal windows and then gridded by means of optimal interpolation before applying the POT method along with the generalized Pareto distribution. Comparison of gridded data with coastal buoy series showed a good agreement, where most of the short-scale

variability associated to the passage of cyclones could be captured. Consequently, the mapping of extreme wave heights return periods showed a good agreement with previous studies and provided more detailed features due to the multi-mission nature of the data and the short-term temporal window. A further improvement of this method should focus on seasonal and monthly periods, as well as the response of local wave climate to the passage of tropical and extra-tropical cyclones. In this sense, a local and regional focus is justified, as the change in the wave climate would not be globally uniform but the mean and maximum

wave heights would increase in middle latitudes and in the Antarctica, along with an increase in the maximum wave heights associated with tropical cyclones (Mori et al., 2009, 2010).

*Author contributions.*  N. da Silva carried out POT computation and data processing, R. de Camargo contribute to the analytical discussion and data analysis, E. Marone contributed to the discussion and edition of the manuscript, H. Sepúlveda collected and processed along-track satellite data. J. Salcedo-Castro prepared the manuscript with contributions from all co-authors. All co-authors reviewed and discussed the

final version of the manuscript.

*Acknowledgements.*  This research is funded by Lloyd's Register Foundation (LRF), which helps to protect life and property by supporting engineering-related education, public engagement and the application of research. Altimeter wave data are made available by Ifremer (ftp://ftp.ifremer.fr/ifremer/cersat/products/swath/altimeters/waves/). We thank the Group of Climate Studies of IAG/USP for providing cyclone tracks information. Doctor H. H. Sepúlveda was supported by the "University of Concepción" fellowship.

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

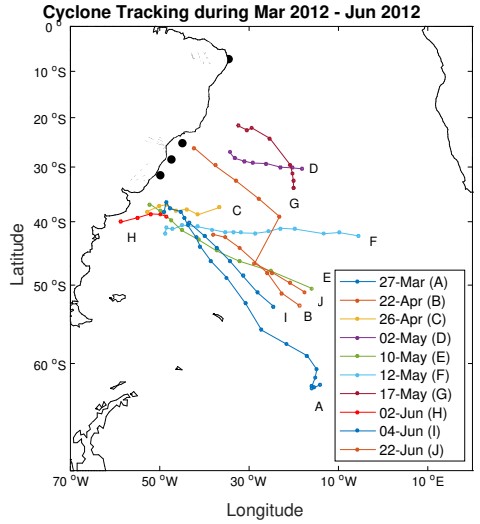

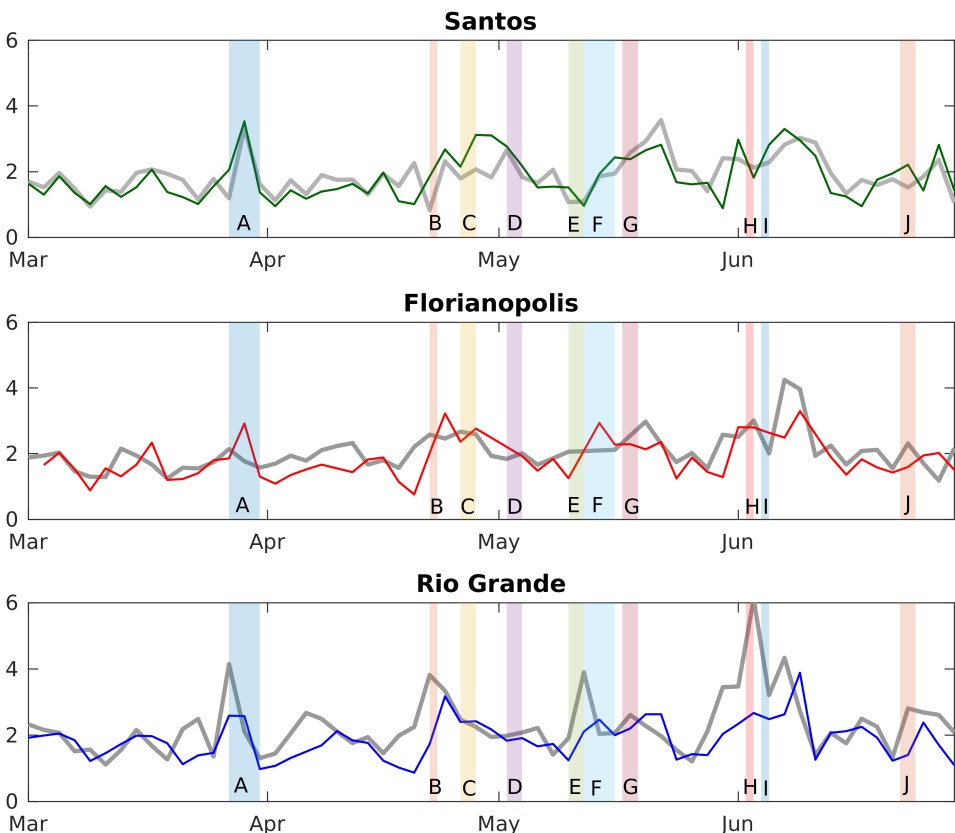

**Figure 5.** (a) Extra-tropical cyclone tracking in the South Atlantic Ocean between March 2012 and June 2012. Buoy stations are marked in black. Initial dates for cyclones occurrences are indicated and each case has a letter assigned (A-J). (b) Time series from linearly interpolated buoy stations (gray) and the nearest correspondent gridded LOS for Santos (green-top), Florianopolis (red-middle) and Rio Grande (blue-bottom). Extra-tropical cyclones occurrences showed in (a) are marked with letters A-J.

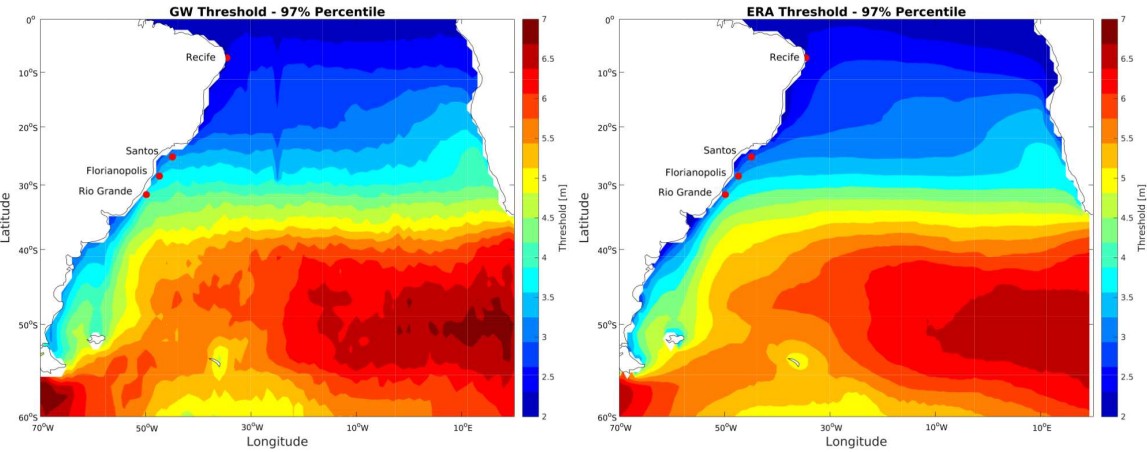

**Figure 6.** Spatial pattern of SWH threshold (97% percentile) values from interpolated LOS data (left) and from ERA-Interim (right) in the South Atlantic.

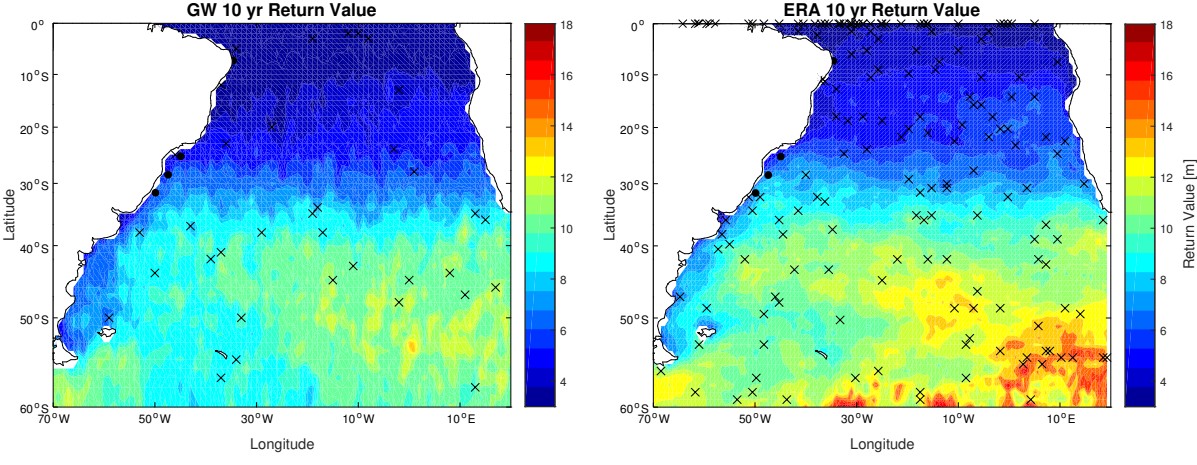

**Figure 7.** Distribution of 10yr return values from interpolated LOS data (left) and from ERA-Interim (right) in the South Atlantic. Crosses represent the series that did not pass the Kolmogorov-Smirnov test.[5]

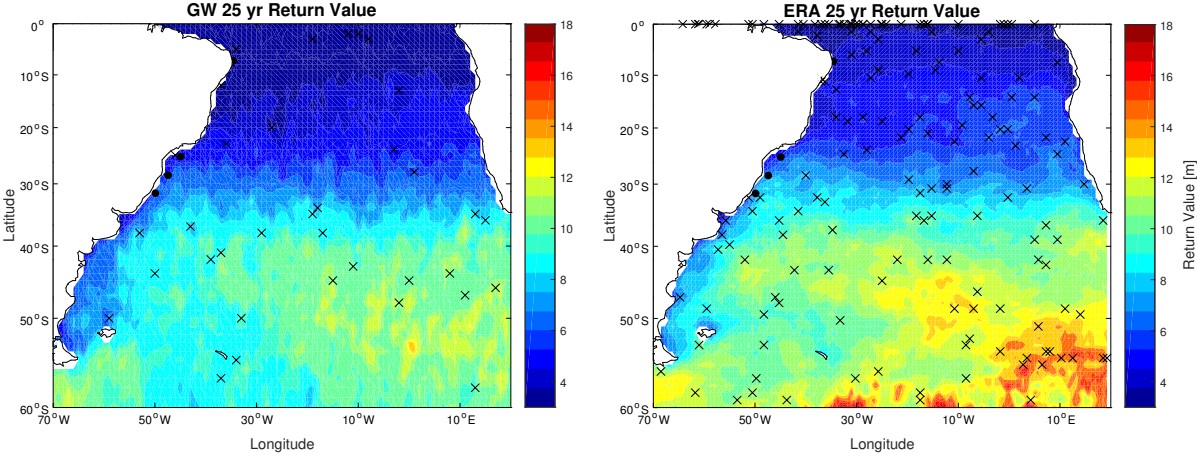

**Figure 8.** Distribution of 25yr return values from interpolated LOS data (left) and from ERA-Interim (right) in the South Atlantic. Crosses represent the series that did not pass the Kolmogorov-Smirnov test.

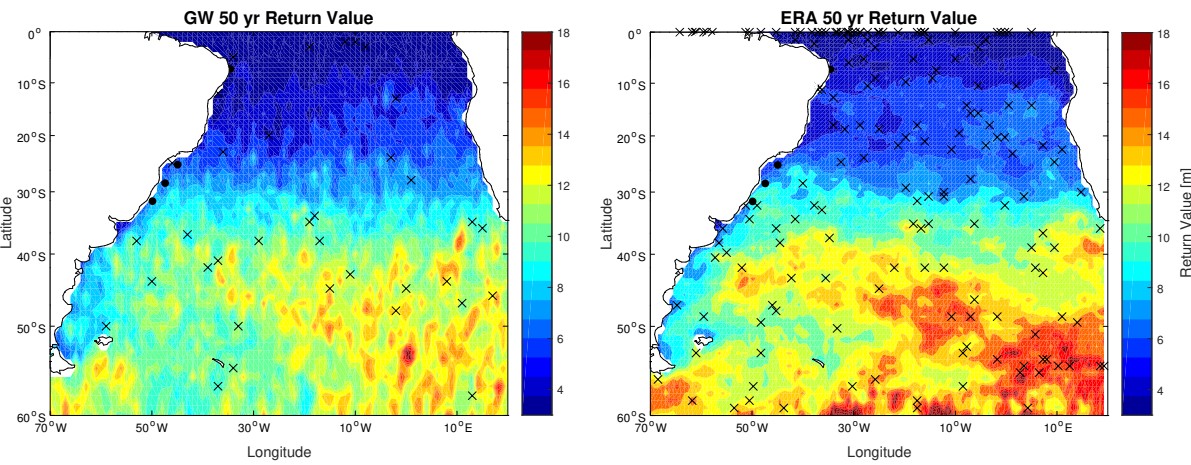

**Figure 9.** Distribution of 50yr return values from interpolated LOS data (left) and from ERA-Interim (right) in the South Atlantic. Crosses represent the series that did not pass the Kolmogorov-Smirnov test.