# Peer review of "Estimation of extreme wave heights return period from short-term interpolation of multi-mission satellite data: application to the South Atlantic"

_Ocean Science, 2017_

## Referee Comment (RC1) · Anonymous Referee #1 · 15 Feb 2018

The paper "Estimation of extreme wave heights return period from short-term interpolation of multi-mission satellite data: application to the South Atlantic" by Julio Salcedo-Castro et al. addresses a vital problem of extracting more reliable information on extreme events from multi-mission satellite altimetry. The paper is reasonably wellorganized and presented in a clear manner, but it should be proof-read, and the English language corrected before the publication. The paper presents a new way of merging satellite data using optimal interpolation (objective mapping) technique. However, the scientific approach is lacking integrity, and quite often the conclusions are

based on insufficiently tested methods. The main concerns are using the data with filtered extremes for the analysis, lack of sufficient justification of the optimal interpolation method, estimation of the threshold data on the basis of in-situ data (not satellite data which were used in the analysis), and insufficient in-situ validation.

The paper is not suitable for publication in Ocean Science in its current form. The main concerns are listed below:

1. The problem with the pre-filtering of the extreme data in the Globwave data processing pipeline still remains. In the technical documentation on the Globwave data (ftp://ftp.ifremer.fr/ifremer/cersat/products/swath/altimeters/waves/documentation/altimeter wave merge 11.4.pdf, page 5) it is described that for each along track sample, a running window of 100 km was applied and within each window, the highest values (approximately 10% of the data) were discarded. Hypothetically, if one can observe a large scale anomaly in wave height with characteristic spatial scales larger than few hundreds of kilometers, then even after this type of filtering the data will still show an extreme event. However, if the extreme events in wave heights are  $\sim$ 100 km or smaller, then the data will be either filtered out completely or show erroneous results. That means that the data can be in principle be used for some limited studies of the wave extremes, but with extreme caution. The authors showed some overall correspondence between the ERA-interim and satellite altimetry return periods, although the question still remains to what extent the Globwave dataset can be used for extreme value analysis. A detailed analysis of the effect of data filtering on the result should be performed before the paper can be published. For example,

a. Getting simulated wave data or model data with extreme events on various spatial scales, extracting data along with the position of satellite tracks, applying the Globwave filtering technique and demonstrating which extreme events can be reconstructed and which are lost during the processing.

b. Obtaining pre-processed along the track data and post-processed data from Ifremer
and estimating a bias introduced by the Globwave filtering technique on the results of the extreme value analysis. Finding how the bias depends on the spatial extent of the extreme event and characteristic length of the event.

c. Comparing the extreme value distribution extracted from Globwave database with extreme value distributions from nearby in-situ data separating data into the duration of the extreme event and its spatial scale. Looking at the best matches with the in-situ data one can find what kind of extreme events can be extracted from the Globwave data.

2. The manuscript describes an interesting application of a new optimal interpolation (objective mapping) technique to satellite altimetry data, and it is written in the abstract and the conclusions section that this is one of the main objectives of the paper. The method is interesting and promising. However, a much more excessive analysis should be performed to test whether this type of interpolation improves the results compared to the previous techniques.

a. The method should be explained in much more details, what exactly is performed during the interpolation, what software package was used (including the version of the package).

b. The authors selected a mesh size of 2x2 degrees based on the estimates of other authors. However, the optimal mesh size found by other authors was found based on different analysis techniques and maybe not correct if additional steps are added in the analysis, such as the optimal interpolation. The optimal mesh size should be found specifically for this method following the techniques of the mentioned in the paper other authors, such as Vinoth and Young 2011 and Wilkin et al. 2002.

c. For each grid point, it should be checked how different are the results of the optimal interpolation and actual data at these points to investigate a bias introduced by the optimal interpolation. Also, the dependence of this bias on the mesh size should be checked. An ultimate test for data corruption by the optimal interpolation would be

OSD
to remove  $\sim$ 20% of the grid points with satellite data, apply the optimal interpolation, restore the data at the locations of discarded points and see how different the values are from the actual measurements.

d. Figure 4 shows differences between the in-situ buoy data and interpolated satellite data. I would like to see non-interpolated data added to that plot and calculated biases and all parameters for non-interpolated data as well so that a reader can see that the optimal interpolation shows a better correspondence to the in-situ data.

e. The authors claim that the optimal interpolation results in a more detailed map (p.11, 110) compared to ERA-interim. It is important either find some additional data to show that these details are real or tell the reader that these details should be treated with caution. It should be proven with additional analysis that the optimal interpolation resolves more real structures in the data.

3. The authors applied a peak over the threshold method to the data; one of the main concerns is that all the tests for finding threshold were performed on in-situ buoy data and not on the actual data used in the analysis. I would like to see the results of the mean excess plot to the satellite altimetry data next to the buoy locations (interpolated and non-interpolated). An application of a different threshold can significantly affect the results of the manuscript.

4. The authors used just four in-situ measurement sites for the validation of the method with quite a short period of observations. There are more in-situ data available for that purpose, for example, the buoys used for the validation of Globwave dataset. I would recommend to get more in-situ data and validate the optimal interpolation method at larger scales so that the proposed method can be trusted not only next to the Brazilian coast, but for the whole South Atlantic.

Detailed comments:

1. p4, l3: "... we firstly added all satellite tracks occurring during two-day temporal win-
dow.." please change the wording, the word "added" is confusing, it might be confused with the addition of all values of satellite tracks.

2. Figure 1: the right picture shows the results of interpolation overlapped with the land. Is it correct?

3. P5, I1: "root-mean-square" -> root mean square

4. P5, I9: "QQplot" -> "quantile-quantile plot"

5. Equation 5: the minus symbol should be outside the fraction, comma should be placed in the equation line, not at the start of the text

6. P8, Figure 3: The dotted lines are not explained in the caption

7. P8, Figure 3, also Table 1: there is an inconsistency between what is written in Table 1 and presented in Figure 3. In Table 1, the Reclife station is said to have data until March 2014 and Santos station data until April 2014. Although, examining Fig.3 the Reclife data are longer. What is the right period of observations?

8. P9, Figure 4: what is the dashed line on the plots? It should be described in the caption

9. P9, Figure 4: The Rio Grande station shows the highest discrepancy between the insitu and satellite interpolated data. The authors argue that the location of the station is the main reason. I would recommend also investigating the fact that the Rio Grande is the closest station to the coast (48km) and that maybe some corruption of the altimeter data by irregularities in the coastline can be responsible for this discrepancy.

10. P11, Figure 6: The black dot showing the Reclife station is not visible in the printed version of the paper. Please consider changing the color of the symbols.

11. P15, I15: "Caires, S: Validation of ocean wind and wave data..." -> "Caires, S. and Sterl, A.: Validation of ocean wind and wave data..."

OSD

---

## Referee Comment (RC2) · Anonymous Referee #2 · 23 Feb 2018

The authors have made a good work to present the spatial pattern of wave extremes in the South Atlantic Ocean by using multiple altimeter data from 1993 to 2015. They used the two-day temporal window and 2âŮę spatial window to get the database and they use that database captured more information than traditional method. However, the paper lacks for details for the reader to be able to fully review these results. The main remarks concern:

(1) The multiple altimeter data is the key for this paper, so more details about the database should be given, such as the accuracy of the database. Which altimeters

have been used to produce the database? And how to make the quality control?

(2) Need to prove that your method is right, nor just compared with other results. Can the buoy data be used to prove your method?

(3) A deeper conclude your results using the tables and figures, and refine your presentations.

(4) Capturing more information on shorter time scales is the characteristic and innovation, give the relationship between the capturing more information and the Estimation of extreme wave heights return period.

In consequence, I think this paper needs major revision.

---

## Author Comment (AC1) · 29 Mar 2018

The authors have made a good work to present the spatial pattern of wave extremes in the South Atlantic Ocean by using multiple altimeter data from 1993 to 2015. They used the two-day temporal window and 2° spatial window to get the database and they use that database captured more information than traditional method. However, the paper lacks for details for the reader to be able to fully review these results. The main remarks concern:
(1) The multiple altimeter data is the key for this paper, so more details about the database should be given, such as the accuracy of the database. Which altimeters have been used to produce the database? And how to make the quality control?
**R**: The description of this aspect was improved in Methods section (P3, lines 25-34). Also, please refer to response to question 1 from Referee #1.

(2) Need to prove that your method is right, nor just compared with other results. Can the buoy data be used to prove your method?
**R**: As described in Results and Discussion section, the comparison of our results with buoy data (Figs. 3 and 4) demonstrate that our method is a valuable approach to capture short term variability.

(3) A deeper conclude your results using the tables and figures, and refine your presentations.
**R**: Conclusions section was improved.

(4) Capturing more information on shorter time scales is the characteristic and innovation, give the relationship between the capturing more information and the Estimation of extreme wave heights return period.
**R**: We agree with the reviewer. A sentence about this point was included in the Conclusions section.

---

## Author Comment (AC2) · 29 Mar 2018

Anonymous Referee #1 The paper "Estimation of extreme wave heights return period from short-term interpolation of multimission satellite data: application to the South Atlantic" by Julio Salcedo-Castro et al. addresses a vital problem of extracting more reliable information on extreme events from multi-mission satellite altimetry. The paper is reasonably well organized and presented in a clear manner, but it should be proof-read, and the English language corrected before the publication.

[Figure]

The paper presents a new way of merging satellite data using optimal interpolation (objective mapping) technique. However, the scientfic approach is lacking integrity, and quite often the conclusions are based on insufficiently tested methods. The main concerns are using the data with filtered extremes for the analysis, lack of sufficient justfication of the optimal interpolation method, estimation of the threshold data on the basis of in-situ data (not satellite data which were used in the analysis), and insufficient in-situ validation. The paper is not suitable for publication in Ocean Science in its current form. The main concerns are listed below:

1. The problem with the pre-filtering of the extreme data in the Globwave data processing pipeline still remains. In the technical documentation on the Globwave data (ftp://ftp.ifremer.fr/ifremer/cersat/products/swath/altimeters/waves/documentation/altimeterpage 5) it is described that for each along track sample, a running window of 100 km was applied and within each window, the highest values (approximately 10% of the data) were discarded. Hypothetically, if one can observe a large scale anomaly in wave height with characteristic spatial scales larger than few hundreds of kilometers, then even after this type of filtering the data will still show an extreme event. However, if the extreme events in wave heights are _100 km or smaller, then the data will be either filtered out completely or show erroneous results. That means that the data can be in principle be used for some limited studies of the wave extremes, but with extreme caution. The authors showed some overall correspondence between the ERA-interim and satellite altimetry return periods, although the question still remains to what extent the Globwave dataset can be used for extreme value analysis. A detailed analysis of the effect of data filtering on the result should be performed before the paper can be published. For example, a. Getting simulated wave data or model data with extreme events on various spatial scales, extracting data along with the position of satellite tracks, applying the Globwave filtering technique and demonstrating which extreme events can be reconstructed and which are lost during the processing. b. Obtaining pre-processed along the track data and post-processed data from Ifremer and estimating a bias introduced by the Globwave filtering technique on the results

[Figure]

of the extreme value analysis. Finding how the bias depends on the spatial extent of the extreme event and characteristic length of the event. c. Comparing the extreme value distribution extracted from Globwave database with extreme value distributions from nearby in-situ data separating data into the duration of the extreme event and its spatial scale. Looking at the best matches with the in-situ data one can iňĄnd what kind of extreme events can be extracted from the Globwave data. R: We understand the concern of the reviewer about the useulness of GlobWave data to study extreme events. In this sense, we inquired GlobWave researchers about this point and the response was that this database should be useful in most cases. Besides, we recognize this point, as cited in our manuscript, (lines 29-32) "some criticism has arisen from some authors, arguing that this processing causes removal of extreme observations and therefore its application to analyze wave extremes is not reliable. However, a recent study by Hanafin et al. (2012) demonstrates that, after correction and processing, this multi-mission altimeter data preserves extreme wave observations larger than 20 m during the passage of extra-tropical cyclones". To address this concern, we examined our data and verified that extreme values are effectively preserved by GlobWave data and, in fact, the storm referred by Hanafin et al. (2012) (14 February 2011) could be detected (See Fig. 1).

Figure 1: Significant wave height (SWH) from GlobWave multi-mission satellite data showing extreme values and the passage of the storm (SHW > 20 m) reported by Hanafin et al. (2012).

Finally, we cite the procedure reported by Queffeulou and Croizé-fillon (2010): "Some other individual spurious measurements (corresponding mainly to high values of SWH) are not explained. Consequently the data are filtered to eliminate these mea- surements. The screening is based on the analysis of the differences between succes- sive along track SWH measurements. For each pass (half orbit) mean value and stan- dard deviation of differences of SWH measurements from pairs of consecutive points are estimated. At 1 second along track sample, two consecutive points are separated

by about 6 or 7 km. A range is then defined by the mean value of the differences plus or minus 3 times the standard deviation (4 times for GFO). Individual data for which the differences with the neighboring measurements are outside this range are then discarded. Specific thresholds are also used at the beginning or at the end of continuous along track series i.e. corresponding to over land passes or to flagged data series. The whole data set from the six altimeter missions were processed in this way. The number of discarded data is low. Only a few measurements per pass, when it happens. All of them appear to be erroneous measurements". In consequence, we are confident that GlobWave data is suitable for extreme wave analysis.

2. The manuscript describes an interesting application of a new optimal interpolation (objective mapping) technique to satellite altimetry data, and it is written in the abstract and the conclusions section that this is one of the main objectives of the paper. The method is interesting and promising. However, a much more excessive analysis should be performed to test whether this type of interpolation improves the results compared to the previous techniques. a. The method should be explained in much more details, what exactly is performed during the interpolation, what software package was used (including the version of the package). R: The optimal interpolation (objective mapping) technique is not new but a standard and proven method in Earth sciences. An improved description of the methods and cited articles is given in the new version of the manuscript, including the software package that was used (Page 4, lines 7-16).

b. The authors selected a mesh size of 2x2 degrees based on the estimates of other authors. However, the optimal mesh size found by other authors was found based on different analysis techniques and maybe not correct if additional steps are added in the analysis, such as the optimal interpolation. The optimal mesh size should be found specifically for this method following the techniques of the mentioned in the paper other authors, such as Vinoth and Young 2011 and Wilkin et al. 2002. R: We made this choice to obtain complete time series (with no gap/missing values) from the widest possible range (1993-2015). Other grids were not tested because this is the

most refined option and, as stated in our manuscript, having both temporal and spatial coverage and resolution is important for our premise (Page 4, lines 17-29).

c. For each grid point, it should be checked how different are the results of the optimal interpolation and actual data at these points to investigate a bias introduced by the optimal interpolation. Also, the dependence of this bias on the mesh size should be checked. An ultimate test for data corruption by the optimal interpolation would be to remove _20% of the grid points with satellite data, apply the optimal interpolation, restore the data at the locations of discarded points and see how different the values are from the actual measurements. R. As the procedure consists in using multiple tracks gathered during 48 hours, this proposal from the reviewers does not seem to provide any relevant estimation. As we stated in point 2.a, the optimal interpolation (objective mapping) technique is not new but a standard and proven method in Earth sciences. Therefore, the application of this method is a routine procedure within the context of the present study.

d. Figure 4 shows differences between the in-situ buoy data and interpolated satellite data. I would like to see non-interpolated data added to that plot and calculated biases and all parameters for non-interpolated data as well so that a reader can see that the optimal interpolation shows a better correspondence to the in-situ data. R. As the previous response, the interpolation method is not relevant part of this study. We understand the concern of the reviewer, however, the focus of this study is the feasibility of using a standard interpolation method to combine multi-mission satellite data in the estimation of extreme wave heights return period. As we assert in the manuscript, unlike smoothing methods, like spline interpolation, objective analysis (or optimal interpolation) is based on the data ensemble statistics, by applying the Gauss-Markov theorem. In this sense, the objective is to assure a result where there is a minimum variance solution at each point. Provided there is a good coverage and knowledge of the data, this method yields quite accurate interpolated results (Betherton, 1976).

e. The authors claim that the optimal interpolation results in a more detailed map

(p.11, l10) compared to ERA-interim. It is important either fi̧nd some additional data to show that these details are real or tell the reader that these details should be treated with caution. It should be proven with additional analysis that the optimal interpolation resolves more real structures in the data. R: We agree with the reviewer. We added a caution sentence in this respect (P10, lines 1-2).

3. The authors applied a peak over the threshold method to the data; one of the main concerns is that all the tests for fi̧nding threshold were performed on in-situ buoy data and not on the actual data used in the analysis. I would like to see the results of the mean excess plot to the satellite altimetry data next to the buoy locations (interpolated and non-interpolated). An application of a different threshold can signifi̧cantly affect the results of the manuscript. R: The reviewer claims that the MEP plots were made with buoy data. That is not true but it was not specified in the text. We included a sentence to clarify this at the beginning of the Results and discussion section (P7, lines 14-16).

4. The authors used just four in-situ measurement sites for the validation of the method with quite a short period of observations. There are more in-situ data available for that purpose, for example, the buoys used for the validation of Globwave dataset. I would recommend to get more in-situ data and validate the optimal interpolation method at larger scales so that the proposed method can be trusted not only next to the Brazilian coast, but for the whole South Atlantic. R: We agree with the reviewer that this is a valid point, but we are focused on the effects of extreme events on Brazil coastal areas (we dedicated some of our results section to compare how the buoy stations and the gw interpolated data responded to the influence of extra-tropical cyclone occurrences). However, we are confident that our results can be expanded and improved in future studies, covering the whole South Atlantic or even globally.

Detailed comments: 1. p4, l3: ": : : we fi̧rstly added all satellite tracks occurring during two-day temporal window.." please change the wording, the word "added" is confusing, it might be confused with the addition of all values of satellite tracks. R: We

replaced the word "added". The word "gathered" seems to be more suitable.

2. Figure 1: the right picture shows the results of interpolation overlapped with the land. Is it correct? R: This figure was modified and improved.

3. P5, l1: "root-mean-square" -> root mean square R: Corrected

4. P5, l9: "QQplot" -> "quantile-quantile plot" R: Corrected

5. Equation 5: the minus symbol should be outside the fraction, comma should be placed in the equation line, not at the start of the text R: Corrected

6. P8, Figure 3: The dotted lines are not explained in the caption R: Dotted lines were part of the graph gridding. These were replaced by lines.

7. P8, Figure 3, also Table 1: there is an inconsistency between what is written in Table 1 and presented in Figure 3. In Table 1, the Recife station is said to have data until March 2014 and Santos station data until April 2014. Although, examining Fig.3 the Recife data are longer. What is the right period of observations? R: Figure 3 only shows an interval where all buoys have most of the data. A longer interval would show several periods with no data and we considered that this would cause noise. However, all calculations were performed considering all buoys data. 8. P9, Figure 4: what is the dashed line on the plots? It should be described in the caption R: This was a graphic artifact that was removed in the current version.

9. P9, Figure 4: The Rio Grande station shows the highest discrepancy between the in situ and satellite interpolated data. The authors argue that the location of the station is the main reason. I would recommend also investigating the fact that the Rio Grande is the closest station to the coast (48km) and that maybe some corruption of the altimeter data by irregularities in the coastline can be responsible for this discrepancy. R: We agree with the reviewer. This sentence was included in the manuscript (P7, lines 25-27).

10. P11, Figure 6: The black dot showing the Recife station is not visible in the printed

version of the paper. Please consider changing the color of the symbols. R: This was a graphic artifact that was removed in the current version.

11. P15, l15: "Caires, S: Validation of ocean wind and wave data: : :" -> "Caires, S. and Sterl, A.: Validation of ocean wind and wave data: : :" R: Corrected

 

Please also note the supplement to this comment:
https://www.ocean-sci-discuss.net/os-2017-81/os-2017-81-AC2-supplement.pdf

[Figure]

[Figure]

**Fig. 1.** Fig. 1 response to reviewer 1

---

## Referee Report (RR1)

Review of

**Estimation of extreme wave heights return period from short-term interpolation of multi-mission satellite data: application to the South Atlantic**

by J. Salcedo-Castro et al.

**Recommendation**

**Major revisions.**

**Synopsis**

The paper describes a method to estimate long-term (10, 25, and 50 years) wave height return values from altimeter data. Along-track SWH values are interpolated in $2 \times 2°$ grid boxes. The results are validated against in-situ (buoy) measurements and compared to estimates based on ERA-Interim data.

**Discussion**

This is a revised version, and I have not reviewed the original version. I therefore concentrate on the remarks of the original reviewer #2. That reviewer mainly asked for additional information. Although the authors state that they have improved the manuscript in reaction to remarks 1, 3, and 4, I cannot see much of additional information, especially not for remarks 3 and 4.

In section 3 a comparison is made between return values derived from the altimeter data and from ERA-Interim (Figs. 7-9). While the *patterns* found from both products are very similar, the altimeter-derived values are lower than those from ERA-Interim, especially in the south-western part of the domain where the differences reach several metres. That's quite a lot. The authors shortly mention this difference, but make no attempt to explain it. Given the size of the discrepancy this is unacceptable. Could the altimeter data be of insufficient quality (or spatio-temporal density) to obtain reliable estimates of return values? What if you sub-sample the ERA-Interim data at the altimeter measurements, and then process and analyse them in the same way as the altimeter data?

**Detailed comments**

Page and line numbers refer to the manuscript version with track-changes.

**p 3, l 21** Croisé-fillon → Croisé-Fillon (also in References)

**p 3, l 32** swh → SWH

**p 12, lines 29-32** Why do you expect these geographical pattern of changes? And what is the relation of this statement with the preceding sentence about seasonal and monthly periods?